# Inflammatory Bowel Disease Therapies and Acute Liver Injury

**DOI:** 10.3390/toxics12060421

**Published:** 2024-06-08

**Authors:** Roberto Catanzaro, Francesco Marotta, Azam Yazdani, Morena Sciuto

**Affiliations:** 1Department of Clinical and Experimental Medicine, Gastroenterology Section, “Gaspare Rodolico” Policlinico Hospital, University of Catania, 95100 Catania, Italy; 2ReGenera R&D International for Aging Intervention, 20144 Milan, Italy; fmarchimede@libero.it; 3Department of Anesthesiology, Perioperative and Pain Medicine, Brigham and Women’s Hospital, Harvard Medical School, Boston, MA 02116, USA; a.mandana.yazdani@gmail.com; 4Specialization School in Digestive System Diseases, University of Palermo, 90133 Palermo, Italy; morenasciuto91@gmail.com

**Keywords:** drug-induced liver injury, cholestasis, hepatotoxicity, liver failure

## Abstract

Drug-induced liver disease (DILI) represents one of the main problems in the therapeutic field. There are several non-modifiable risk factors, such as age and sex, and all drugs can cause hepatotoxicity of varying degrees, including those for the treatment of inflammatory bowel diseases (IBD). The aim of this review is to illustrate the adverse effects on the liver of the various drugs used in the treatment of IBD, highlighting which drugs are safest to use based on current knowledge. The mechanism by which drugs cause hepatotoxicity is not fully understood. A possible cause is represented by the formation of toxic metabolites, which in some patients may be increased due to alterations in the enzymatic apparatus involved in drug metabolism. Various studies have shown that the drugs that can most frequently cause hepatotoxicity are immunosuppressants, while mesalazine and biological drugs are, for the most part, less associated with such complications. Therefore, it is possible to assume that in the future, biological therapies could become the first line for the treatment of IBD.

## 1. Introduction

Inflammatory bowel diseases (IBD) are relatively widespread intestinal pathologies with a constantly increasing incidence. Over the decades, several studies have been conducted aimed at developing new pharmacological therapies for these pathologies. To date, commonly used drugs include aminosalicylates, corticosteroids, immunosuppressants, biologics and immunomodulators [1]. In patients with IBD, anomalies in the biochemical parameters of liver function can be found in percentages ranging from 3% to 50% of cases. The liver diseases that can develop in these cases include hepatic steatosis, primary sclerosing cholangitis (PSC), cholelithiasis, autoimmune hepatitis, cirrhosis, cholangiocarcinoma and drug-induced liver injury (DILI) [2,3].

Today, DILI represents the most common cause of acute liver failure (ALF). It is characterized by an increase in transaminases as much as five times the normal value, the appearance of jaundice and liver-related coagulopathy [4]. DILI can be direct, indirect or idiosyncratic (Figure 1), (Table 1).

The first case occurs when direct hepatotoxicity occurs from agents intrinsically toxic to the liver. The indirect form is due to the action of the drug and not to its toxic formulation. These two forms are typically dose-dependent. Idiosyncratic DILI is associated with minimally or not at all toxic drugs and is, therefore, most likely related to the metabolic characteristics of the liver itself. The latter type of DILI appears to be non-dose-dependent [5].

In direct and indirect liver injury, increases in transaminases and/or alkaline phosphatase are generally observed without hyperbilirubinemia. In these cases, symptoms may be mild or even absent [6]. Serum levels of liver function tests decrease when the drug is stopped or the dose is reduced. In some cases, these alterations can also be transient and resolve spontaneously without stopping the drug as an adaptation occurs. If this does not happen and the drug is continued, the damage progresses and jaundice and symptoms may also appear [5]. In severe cases of direct liver injury, acute hepatic necrosis occurs. In particular, a histological examination will reveal a centrilobular or panlobular necrosis with mild inflammation, a pattern similar to that of ischemic hepatitis. This condition can also be fatal [7].

The idiosyncratic form is classified into three groups based on serum alanine aminotransferase (ALT) and alkaline phosphatase (AP) values: “acute hepatocellular hepatitis” if ALT > 2 N or by an ALT⁄AP ≥ 5; “acute cholestatic hepatitis”, with isolated elevation of AP > 2 N or an ALT⁄AP ≤ 2, and “mixed pattern”, defined by simultaneous presentation of both ALT > 2 N and elevation of AP and ALT/AP ratio between 2 and 5 [8].

Idiosyncratic damage associated with acute hepatocellular hepatitis is similar to acute hepatitis of viral origin. Conspicuous increases in transaminases will be observed, while alkaline phosphatase tends to increase to a lesser extent [9].

Acute cholestatic hepatitis is characterized by prominent symptoms of pruritus, jaundice, and moderate-to-severe elevations in AP levels. This form is usually self-limiting. At the histological level, lesions of the bile duct can be observed in cholestasis in the small bile canaliculi [10].

Finally, the idiosyncratic mixed form is determined by substances that cause both hepatocellular damage and cholestasis. This is the type of damage with better outcomes, which rarely leads to liver failure [11].

Over the last few years, any factors that may be associated with a greater susceptibility towards the development of DILI have been evaluated. For example, some studies have highlighted a higher prevalence of DILI in women (59% women versus 41% men). This is because, for example, there are important differences in various aspects of the pharmacokinetics or pharmacodynamics of drugs between men and women. But not only that, hormones or the immune system can also influence reactions to some drugs [12].

However, the elderly have an almost threefold increase in the incidence of DILI. This has been attributed to the high amount of drugs taken by elderly patients [13].

Another factor that can influence a hepatic adverse drug event is represented by the gut microbiota. The gut microbiota, in fact, also has an effect on the physiology of the liver. In particular, the translocation of intestinal bacteria occurs via the portal circulation, which can, therefore, also be infiltrated by pathogenic microorganisms and their products [14].

The bacteria present in the intestine can, in fact, interfere with the metabolism of drugs, and it can, in certain cases, cause an increase in the toxicity of these molecules even at therapeutic doses [15].

For example, it emerged that the *Firmicutes*/*Bacteroidetes* ratio is high in subjects who develop DILI, suggesting a probable pathogenetic role of Firmicutes. Instead, other species, such as those of the genus *Lactobacillus*, were found to be hepatoprotective [16].

Indirect evidence of the influence determined by the gut microbiota is represented by the effectiveness of some probiotics in reducing liver damage. This action is carried out both by maintaining the integrity of the intestinal barrier and by stimulating the secretion of short-chain fatty acids (SCFA) [17].

Unfortunately, to date, there are no specific biomarkers that can allow early identification of subjects who could develop DILI following the intake of a specific drug, regardless of the dose. MicroRNAs (miRNAs) are small non-coding RNAs involved in post-transcriptional regulation of gene expression. They have significant importance in the pathogenesis of diseases since their expression changes during the evolution of disease-causing organ damage [18]. In this case, in fact, an increase in miRNA levels was seen in both bloodstream and urine. MiRNAs are relatively stable in biofluids, a feature that has contributed to circulating miRNAs that have received much attention lately as potential non-invasive DILI biomarker candidates. To date, several studies have reported changes in serum miRNA concentrations during liver injury [19].

Therefore, it is reasonable to think that before long, these biomarkers could be used in the early evaluation of drug-related hepatotoxicity.

## 2. Aminosalicylates

Aminosalicylates include sulfasalazine and its catabolites, sulfapyridine and 5-aminosalicylic acid (5-ASA), also known as mesalazine. The use of sulfapyridine has been progressively reduced due to the severe associated adverse effects, including acute hepatitis [20]. In fact, sulfasalazine contains both 5-aminosalicylic acid and sulfapyridine and the latter molecule is responsible for the toxic effects. In particular, it causes a hepatic hypersensitivity reaction, which can evolve into various forms of liver damage: granulomatous hepatitis, cholestatic liver disease and, in rare cases, even acute liver failure [21]. A study conducted in mice identified several pathways involved in sulfasalazine-induced liver damage. These include redox processes, the cytochrome p450 pathway, glutathione metabolism and the cytochrome p450 2C55 pathway [cyp2c55] [22].

Mesalazine is the aminosalicylate most commonly prescribed to patients with ulcerative colitis (UC) or Crohn’s disease for the maintenance of remission and in association with corticosteroids for the induction of remission in mild-moderate active forms. The incidence of mesalazine-associated hepatotoxicity remains low because mesalazine is minimally absorbed and mostly eliminated in the feces [23,24]. Adverse hepatic effects associated with mesalazine range from a mild asymptomatic increase in hepatic cytolysis rates to idiosyncratic cholestasis, and in most cases, these events resolve rapidly with discontinuation of the drug [25]. In 0–4%, an increase in liver enzymes was detected. In a randomized controlled trial, it emerged that 1% of patients taking the lowest daily dosage of mesalazine (1.5 g/day) experienced hypertransaminasemia, which resolved after discontinuation of the drug. Then, upon resumption of treatment 4 weeks later, liver cytolysis enzymes increased again [22,26].

Serious hepatic complications from 5-ASA are extremely rare in the literature to date. A recent case report by Watanabe et al. presented the case of a patient with UC who, on the 98th day of therapy with mesalazine at a dosage of 2.4 g/day, developed adverse events. In particular, an increase > 5 times the norm of transaminases and a significant increase in cholestasis indices were found. Once 5-ASA was discontinued and all other possible causes of DILI were excluded, it was possible to define 5-ASA-induced DILI, and after 24 days from drug suspension, a normalization of all liver enzymes was observed [27].

Several studies suggest monitoring liver function tests before and during treatment with sulfasalazine or mesalazine to diagnose any acute events early [3].

## 3. Immunosuppressants

Thiopurines used in the treatment of IBD include azathioprine (AZA) and 6-mercaptopurine (6-MP). These molecules can cause dose-independent reactions, therefore allergic or idiosyncratic, or dose-dependent adverse effects. Hepatotoxicity can be represented by acute hepatocellular damage, in which transaminases mainly increase, or by acute cholestatic hepatitis, with a prevalent increase in cholestasis indices [28].

Azathioprine is used in the treatment of IBD, particularly Crohn’s disease. It has been associated with several forms of hepatotoxicity. This drug is a purine analog that interferes with the cell cycle and inhibits the normal function of leukocytes. In this way, its immunosuppressive action is implemented. 6-MP is the active metabolite of azathioprine and is further metabolized to active metabolites, including 6-methylmercaptopurine, thioguanine, 6-thioguanine nucleosides and 6-methylmercaptopurine nucleosides [29]. Among the adverse events of these drugs is hepatotoxicity, which can occur in approximately 10% of patients. It does not appear to be dose-related, as it can also be observed in patients with low concentrations of 6-methylmercaptopurine [30].

An important step in AZA metabolism includes the involvement of glutathione in hepatocytes, with the conversion of AZA to 6-MP and methylnitroimidazole by glutathione S-transferase. Therefore, in patients in whom there is high activity of the hepatic glutathione S-transferase enzyme, there is an increased risk of hepatotoxicity induced by the excessive release of methyl-nitroimidazole and MP [31]. The various expressions of hepatotoxicity caused by this drug vary from simple asymptomatic increases in transaminases to real forms of acute hepatitis, which can be mainly cholestatic or mixed. Vascular endothelial lesions, peliosis hepatis and sinusoidal syndrome may also be observed expansion [32]. In a study of nearly 4000 IBD patients, Chaparro et al. have highlighted that hepatotoxicity is one of the most common adverse events of thiopurines, with a frequency of 4% [33].

The high incidence of hepatotoxicity of thiopurines is also demonstrated by the higher rates of dose reduction or interruption of therapy compared to those of other drugs used in IBD. Additionally, patients treated with 6-MP had higher dose reduction rates than those treated with AZA, although discontinuation rates were similar in the two groups [34]. Thiopurine-associated liver damage has been related to the activity of thiopurine S-methyltransferase (TPMT), an enzyme involved in the metabolism of both 6-MP and azathioprine [3,25]. In fact, the presence of genetic polymorphisms can decrease the activity level of this enzyme resulting in variable levels of thiopurine metabolites influencing the degree of hepatotoxicity [35].

In several cases, in the early stages of treatment, minimal and transient increases in transaminases can be observed without liver damage. In other cases, azathioprine can also cause the onset of acute liver damage of a cholestatic type, which manifests itself with jaundice, fatigue, increased transaminases and alkaline phosphatase. This pathological condition usually resolves with discontinuation of azathioprine [36].

The mechanism of hepatotoxicity induced by this drug is not fully known. Azathioprine is a prodrug that is metabolized into 6-MP. Subsequently, 6-MP can undergo three different metabolic pathways. The first consists of the methylation of 6-MP into 6-methylmercaptopurine (6-MMP), a reaction catalyzed by thiopurine methyltransferase (TPMT). The second pathway involves the conversion of 6-MP to 6-thioinosine 5-monophosphate via hypoxanthine guanine phosphoribosyl transferase, and this intermediate is then metabolized in the active nucleotides 6-thioguanine (6-TG). The third pathway consists of the transformation of 6-MP into 6-thiouric acid (6-TA), which is an inactive metabolite, by xanthine oxidase (XO) (Figure 2) [37]. It has been observed that some subjects tend to produce more 6-MMP rather than 6-TG, and they are more likely to develop hepatotoxicity [38]. The conversion of 6-MP to 6-TA can also be hepatotoxic, as this metabolic pathway is a source of reactive oxygen species (ROS) [37]. In addition to the characteristics of the enzymatic apparatus of each individual, the simultaneous administration of other drugs can also facilitate DILI due to AZA. For example, allopurinol works by inhibiting xanthine oxidase, and this, therefore, results in an increase in 6-MP [39].

Methotrexate (MTX) is a competitive inhibitor of the enzyme dihydrofolate reductase, which participates in the synthesis of purines and pyrimidines, producing antiproliferative and anti-inflammatory effects on cells. It is used in various chronic pathologies with inflammatory pathogenesis, such as rheumatoid arthritis, psoriatic arthritis and also IBD. In fact, it aims to induce and maintain remission in these types of patients [40]. However, this drug is burdened by several adverse effects that can affect various organs, including the liver. In particular, during long-term therapies with MTX, the accumulation of the polyglutamate metabolite of MTX occurs in the liver cells which, most likely, is responsible for the toxic effects at the liver level [41]. In some cases, liver damage caused by methotrexate manifests itself with a transient alteration of liver function indices, which return to normal despite continuing therapy, without dose changes or without suspension of the drug [42].

It should be noted that liver damage from MTX occurs more in patients with IBD than in patients with other pathologies. For example, in a study by Fegan et al., it emerged that 17.5% of IBD patients treated with MTX had increased serum aminotransferases [43]. Even Fournier et al. found that 24% of IBD patients developed abnormal liver function with MTX [44]. A similar result was obtained by González-Lama et al., in which at least 20.8% of this type of patients presented liver function abnormalities or even significant liver fibrosis [45]. On the contrary, in various studies, the percentage of liver function abnormalities was found to be lower in patients treated with MTX with non-IBD diseases. For example, Lie et al. found an increase in liver enzymes in patients with rheumatoid arthritis (RA) of less than 10% [46].

These peculiar differences can be explained by the mechanism of action of the drug. MTX directly inhibits various enzymes involved in folate metabolism, including the dihydrofolate reductase (DHFR) enzyme. Inhibition of the latter can lead to the blocking of DNA replication and consequent cell death [47]. In particular, the rs1650697 (C35T) polymorphism was discovered, which concerns the promoter of the DHFR gene, which appears to have a protective effect against the hepatotoxicity of MTX. This polymorphism appears to be more frequent in subjects with RA than in those with IBD [48,49].

Other genetic polymorphisms that could be responsible for protective or facilitatory effects against the hepatotoxicity of MTX concern another gene that codes for the methylene tetrahydrofolate reductase (MTHFR) enzyme. This is the A1298C polymorphism, which has been found much more frequently in patients with IBD and appears to be associated with a greater hepatotoxic effect of MTHFR [41]. Various risk factors have been identified, including alcohol intake, obesity, diabetes mellitus and chronic viral hepatitis. Instead, the intake of folic acid proved to be protective in terms of reducing the incidence of adverse events from MTX. It is, therefore, important to recommend that patients who must undertake this therapy avoid alcohol intake and instead integrate folic acid [28].

Cyclosporin A (CsA) is a calcineurin inhibitor used to induce remission in cases of severe ulcerative colitis refractory to steroids [50]. It has been known for some time that CsA can cause DILI. At the serum level, an increase in transaminases, indices of cholestasis, bilirubin and also an increased production of bile acids can be found. Instead, histological alterations induced by CsA include congestion and dilatation of the bile ducts, activation of Kupffer cells, infiltration of inflammatory cells into the interstitium and focal necrosis of hepatocytes [51]. The mechanisms by which CsA is hepatotoxic are different. Certainly, among these, there is oxidative stress. In fact, CsA increases the production of reactive oxygen species (ROS), and it inhibits the Krebs cycle and oxidative phosphorylation, with consequent reduction of adenosine triphosphate (ATP) production at the mitochondrial level. Furthermore, there would also appear to be a reduction in intracellular antioxidant systems [52].

Due to the high percentage of adverse effects of CsA (not only at the liver level), this drug is now rarely used in the therapy of IBD.

Tacrolimus is an immunosuppressant primarily used in the prevention of organ transplant rejection but is also used in the treatment of patients with corticosteroid-refractory UC and refractory perianal fistulizing CD [53]. Tacrolimus binds to a binding protein in T cells, forming a new complex molecule that binds and inhibits calcineurin. This blocks the production of T cell-derived cytokines, such as IL-2–IL-7, interferon-γ and TNF. In this way, inflammation is counteracted. However, it can also lead to the production of ROS, causing cell death by apoptosis [54]. These effects are almost always dose-dependent; therefore, this drug should only be used short-term for induction of IBD remission [53].

## 4. Biologic Therapies

### 4.1. Tumor Necrosis Factor Inhibitors

Tumor necrosis factor (TNF) is released by T-cells and macrophages and binds to specific receptors that induce multiple immune responses, including the release of inflammatory cytokines and the migration of leukocytes to organs. TNF inhibitors are Infliximab, Adalimumab, Golimumab and Certolizumab pegol. They antagonize TNF receptors and induce inhibition of immune responses by TNF, thus inducing recovery and maintenance of remission of CD and UC [55]. Generally, the most common manifestation of anti-TNF hepatotoxicity is acute hepatocellular damage, which occurs on average 13 weeks after the start of therapy. More rarely, mild cholestasis has also been observed. The mechanism by which these drugs cause hepatotoxicity is, however, unknown [56].

Infliximab is the first chimeric monoclonal antibody working against produced TNF. It is usually dosed at 5 mg/kg or 10 mg/kg and administered intravenously, with induction therapy at 0, 2, 6 and 8 weeks, followed by maintenance therapy with an infusion every 8 weeks [25]. This drug may cause several adverse effects, including liver damage, which can be represented by a hepatocellular pattern or an autoimmune type [57].

In the autoimmune pattern, positivity for autoantibodies (anti-nucleus antibodies, anti-smooth muscle antibodies, etc.) can be observed, while from a histological point of view, the picture is that of interface hepatitis [58].

Immune-mediated liver injury is the most frequent and severe form of Infliximab-induced liver disease. However, direct hepatotoxicity is generally transient and asymptomatic. Severe hypertransaminasemia is highly rare and occurs primarily in individuals taking other hepatotoxic drugs or who already have liver disease at the time of initiation of treatment with Infliximab [59]. This is the case of a young 25-year-old patient undergoing therapy with Infliximab 400 mg IV every 8 weeks and Methotrexate 10 mg once a week. In fact, two months after starting treatment, the woman developed anorexia and weight loss with subsequent suspension of Methotrexate. A few weeks later, he then developed pruritus, jaundice and asterixis. Laboratory tests and liver biopsy made it possible to diagnose infliximab-related AIH. In this specific case, the patient experienced a progressive deterioration of her clinical conditions, which made an orthotopic liver transplant necessary [60].

Indeed, in a retrospective case-control study of patients with IBD without concomitant liver disease, approximately one-third of patients experienced ALT elevations, which, however, resolved spontaneously in most cases. These biochemical alterations, however, were not significantly associated with the use of infliximab, so it is reasonable to assume that the presence of liver disease prior to the start of therapy with Infliximab could significantly influence the outcome of these patients [61].

Infliximab therapy rarely causes DILI. In most cases, only biochemical alterations are present. In a study by Worland et al. conducted on 157 patients with IBD treated with Infliximab, one-third of them had liver biochemical abnormalities, while only one met the RUCAM criteria for DILI [62]. Infliximab-induced DILI usually develops after several infusions, with an average latency of 14–18 weeks after induction [63].

CT-P13 (Janssen Biotech, Horsham, PA, USA) is a biosimilar drug of anti-TNF alpha Infliximab and has been approved by the European Medicines Agency (EMA) and the US Food and Drug Administration (FDA). Biosimilars do not present significant differences in terms of safety and efficacy compared to the original molecules [64].

Some rare cases of DILI have also been reported for this biosimilar. The first published case involved a 23-year-old woman with CD who developed DILI after switching from the original IFX to the biosimilar IFX CT-P13. A liver biopsy showed pericentral canalicular cholestasis, with no other findings related to steatosis or sclerosing cholangitis. CT-P13 was discontinued, and the patient improved 10 weeks later. The subsequent switch to the original Infliximab did not lead to alterations in liver function indices [65].

A similar event was reported by Zachau et al. about a 42 year old patient suffering from Crohn’s. Also, in his case, the suspension of the biosimilar resulted in an improvement in the clinical picture until normal transaminase values and cholestasis indices were restored, and the subsequent switch to the original Infliximab did not cause the appearance of symptoms or biohumoral alterations [64].

Adalimumab is a human monoclonal anti-TNF-alpha antibody that is administered subcutaneously, usually every 2 weeks, in the maintenance phase of remission. Compared to infliximab, less hepatotoxicity was detected in Adalimumab. Koller et al. observed increases no greater than two times normal in transaminases in 135 treated IBD patients with Adalimumab [66]. In cases of liver disease related to Adalimumab, it was seen that this resolved without relics when the drug was suspended [67].

### 4.2. Anti-Integrin Antibodies

Anti-integrin antibodies are molecules capable of blocking integrins, which are surface proteins involved in the migration of leukocytes in the intestinal mucosa, one of the events responsible for the onset of chronic inflammation in the gut [68]. Integrins are composed of heterodimeric α and β subunits that bind components of cell adhesion molecules (CAMs) and the extracellular matrix. Certain stimuli, such as the presence of cytokines, can induce conformational changes in integrins that increase their affinity for ligands. This results in a migration of lymphocytes within the tissue [69].

Vedolizumab is a humanized antibody directed against α4β7 integrin. It blocks the interaction between α4β7 integrin and mucosal addressin-cell adhesion molecule 1 (MAdCAM-1) expressed on endothelial venules in gut-associated lymphoid tissue (GALT). In this way, it prevents the adhesion of leukocytes to the cells of the intestinal endothelium [70].

Its efficacy has been demonstrated in several clinical studies and is approved for the induction and maintenance of remission of moderate–severe ulcerative colitis (UC) and Crohn’s disease [71]. In particular, the symptoms of the disease reported by the patients progressively reduced already in the remission induction phase and improved in a statistically significant manner compared to the placebo by the end of this phase [72]. This biological drug appears to have a high safety profile as it has a selective action on the intestine, resulting in fewer systemic effects [70]

Some cases of DILI due to vedolizumab have been reported, but it is a transitory condition that ceases once the drug is discontinued [61]. The final analysis of the GEMINI LTS study found that only 3.2% of patients with UC and 4.7% of MC patients developed damage to the liver, and, in any case, in none of these cases was it necessary to suspend Vedolizumab [73].

Natalizumab is a humanized IgG4 monoclonal antibody that works against the α4 subunit of integrin, in particular α4β1 and α4β7. It was approved for the treatment of moderate–severe CD as it proved to be effective and with a high safety profile [70]. It has rarely resulted in cases of liver toxicity. In particular, less than 5% of patients treated with Natalizumab developed slight increases in aminotransferases during therapy, while fulminant liver failure associated with it has been reported in the literature in less than 1% of cases [74]. The alterations in liver function indices were found both after the administration of the first or second dose and after several infusions. Generally, it was non-serious acute hepatitis, which resolved spontaneously after stopping the drug [75].

### 4.3. Anti-Interleukin 12/23 Antibodies

Anti-interleukin 12/23 antibodies include Ustekinumab, a direct human monoclonal antibody against the p40 subunit of interleukin (IL)-12 and IL-23 [76]. It has been approved by the FDA for the treatment of adult patients with moderate–severe IBD, as it has been shown to be effective in inducing and maintaining clinical remission in patients with CD and UC [77].

Ustekinumab was described as safe from a hepatic perspective according to phase III and IV studies and PHOENIX I and II studies. To date, rare cases of autoimmune hepatitis induced by this drug have been detected, but the mechanisms by which Ustekinumab causes this type of adverse effect are not yet known [78].

### 4.4. Janus Kinase Inhibitors

Janus kinase (JAK) inhibitors are drugs that act on a family of intracellular tyrosine kinases capable of transducing cytokine-mediated signals through the STAT pathway. The latter is involved in several biological processes, including inflammatory responses [79]. Therefore, JAK inhibitors are able to block the effects of several types of cytokines, thus determining a more effective therapeutic response compared to TNF-α inhibitors or integrin inhibitors [80].

The first JAK inhibitor approved for the treatment of moderate–severe UC was tofacitinib. Also, for this drug, among the adverse events, an increase in serum aminotransferases has been reported in 28–34% of cases. In these patients, only mild liver involvement was detected [81]. However, in a recent case report, Mardani et al. reported an episode of liver failure with hypertransaminasemia, jaundice and increased indices of cholestasis in a patient receiving tofacitinib (at a dose of 5 mg/daily). The mechanism by which the drug caused liver damage is not known. Production of toxic or immunogenic intermediates may have occurred, but this has not been established [82].

As regards the other two JAK inhibitors currently approved for the treatment of moderate–severe forms of UC, upadacitinib and filgotinib, transient increases in transaminase values have been recorded. To date, no cases of acute hepatocellular damage induced by these drugs have been reported [83].

### 4.5. Sphingosine-1-Phosphate (S1P) Receptor Modulators

Sphingosine-1-phosphate (S1P) receptor-modulating drugs have recently been developed for the treatment of several immune-mediated diseases, including IBD. The S1P receptor intervenes in the regulation of cell proliferation and migration, participates in intercellular communication and also has other effects on the cardiovascular system [84].

Ozanimod is an S1P receptor modulator already approved in the USA for the induction and maintenance of remission in patients with moderate–severe UC and in Europe in patients who did not benefit from or were intolerant to other biological drugs [85]. To date, some cases of patients treated with Ozanimod who showed an increase in liver function indices during the induction or maintenance period have been reported from the True North and Touchstone studies. In particular, an increase in ALT levels above three times the upper limit of normal (ULN) was reported in 2.6% of cases during the induction period and in 2.3% of cases during maintenance. Instead, ALT increases above five times the ULN occurred in 0.9% of cases during induction and 0.9% of cases during maintenance [85,86].

For some of the biologic drugs, there is still limited information regarding adverse effects on the liver, as they have recently been approved for the treatment of IBD. Therefore, future studies will be necessary to illustrate the percentages of adverse effects at a systemic level, not just the liver, so as to highlight the safest drugs, which could, therefore, be set as first-line therapies.

## 5. Conclusions

Drugs used in the treatment of IBD can cause hepatotoxicity and lead to DILI. This review has considered the various studies conducted over the years to evaluate the adverse effects of these drugs on the liver and provides an overview of the safest drugs and those that should no longer be used. In this regard, it has emerged that immunosuppressants are the category of drugs most involved in the genesis of adverse hepatic effects. Instead, drugs that have always been considered last-line, such as biologics, have almost all proven to be much safer than older molecules. The only exception is represented by anti-TNFs, in particular Infliximab, which are still burdened by a non-negligible percentage of patients who develop hepatotoxicity. The other categories, such as anti-integrin antibodies, appear to be better tolerated. Therefore, considering the negligible percentage of adverse effects compared with the high efficacy in terms of induction and maintenance of remission, it is reasonable to assume that biological drugs could become first-line therapies for the treatment of IBD in the not-too-distant future.

## Figures and Tables

**Figure 1 toxics-12-00421-f001:**
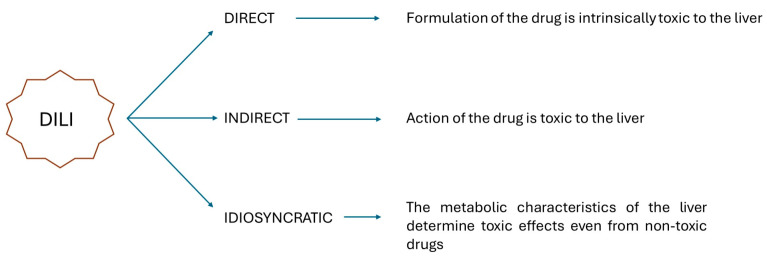
Three types of drug-induced liver injury (DILI).

**Figure 2 toxics-12-00421-f002:**
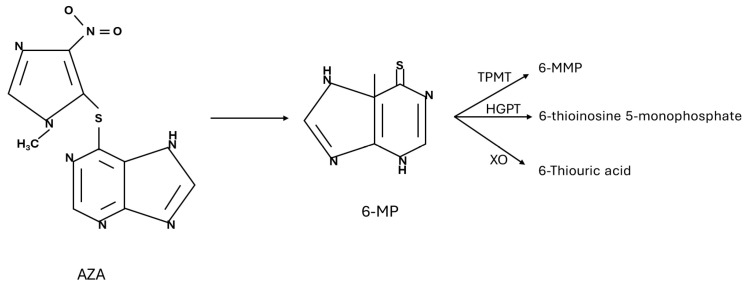
Metabolism of thiopurine drugs (AZA: azathioprine; 6-MP: 6-mercaptopurine; TPMT: thiopurine methyltransferase; HGPT: hypoxanthine guanine phosphoribosyl transferase; XO: xanthine oxidase; 6-MMP: 6-methylmercaptopurine).

**Table 1 toxics-12-00421-t001:** Three types of drug-induced liver injury (DILI) with respective biochemical parameters, symptoms and hystology. AP: alkaline phosphatase; ALT: alanine aminotransferase; 2 N: twice the norm.

DILI Forms	Biohumoral Parameters	Symptoms	Histology
Direct/indirect (mild-moderate)	↑ transaminases and/or AP ± bilirubin	mild/absent	
Direct/indirect (severe)	↑↑ transaminases and/or AP ± bilirubin	jaundice, pruritus	centrilobular or panlobular necrosis with mild inflammation
Idiosyncratic:			
- acute hepatocellular hepatitis	ALT > 2N and/or ALT/AP ≥ 5	mild/absent	cholestasis in the small bile canaliculi
- acute cholestatic hepatitis	AP > 2N and/or ALT/AP ≤ 2	jaundice, pruritus	
- mixed pattern	ALT > 2N, AP > 2 and ALT/AP = 2–5	mild/absent	

## Data Availability

Not applicable.

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
