# Peer review of "Inflammatory Bowel Disease Therapies and Acute Liver Injury"

_toxics, 2024, doi:10.3390/toxics12060421_

Round 1
Reviewer 1 Report
Comments and Suggestions for Authors
Drug-induced liver disease (DILI) is a significant concern within the realm of therapeutics. Like many conditions, it exhibits various identifiable risk factors, including age, sex, and even the gut microbiota. Hence, virtually all medications on the market have the potential to cause hepatotoxicity to varying extents. This risk extends to drugs used in treating inflammatory bowel disease (IBD), which can precipitate adverse hepatic effects, potentially leading to DILI. This review examines the diverse classes of drugs utilized in IBD treatment, elucidating their respective impacts on liver function. Notably, immunosuppressants have emerged as significant contributors to hepatotoxicity, while biological drugs appear to pose a comparatively lower risk.
The content of this article is quite substantial and comprehensive. I believe this article is exceptionally well-written and deserving of publication. However, I have only few comments and questions.
1. Please include the full name alongside the abbreviation in the abstract.
2. In the Conclusions section, it is crucial to provide a detailed explanation of the significant contributions of the study.
3. These two articles need to be cited.World J Hepatol 2021; 13(12): 1828-1849. World J Hepatol. 2021; 13(12): 1956-1967.
Author Response
Dear reviewer, thanks for your suggestions. We have made the changes you requested. In the abstract we have included all full names before abbreviations. We have discussed the "Conclusions" paragraph further. We have cited the two articles you suggested.
Reviewer 2 Report
Comments and Suggestions for Authors
1. Abstract: acronyms that appear for the first time should have full names; There is no overall summary of the content of the full text, and the significance of writing this review is not outstanding.
2. The structure of the review is not clear, the hierarchy is too chaotic, and the links between the upper and lower paragraphs are not tight, like appearing out of thin air.
3. From the conclusion, I guess that the author wants to express that biologic therapies have fewer complications and may become the first choice for the treatment of IBD in the future, but this view is not reflected any in the article.
4. The references cited are not new enough.
There is no problem with English language.
Author Response
Dear reviewer, thanks for your suggestions. We have made the changes you requested.
In the abstract we have included all full names before abbreviations and we have tried to better summarize the various topics covered in the review.
We have tried to improve the structure of the review by creating additional subsections where necessary.
We have almost completely rewritten the "Conclusions" paragraph to try to make it more in line with the content of the text.
We have expanded the number of references by adding more recent articles
Reviewer 3 Report
Comments and Suggestions for Authors
The manuscript “Inflammatory bowel disease therapies and acute liver injury” dealt with the relationship between some inflammatory bowel disease therapies and acute liver injury. The manuscript is of practical significance to the safety of medicines.
1) Abstract of the manuscript need to be rephrased because some sentences seem to be out of the topics, such as “As with most pathologies, also in this case various risk factors can be recognised, such as age, sex, but also the gut microbiota”. Besides, the conclusion of effects of the important drugs on liver should be drawn in Abstract.
2) Figure 1 seems to uninformative.
3) In the section “2. Compounds of 5-aminosalicylic acid”, is 5-aminosalicylic acid one type of compound?
To sum up, major revision is needed.
Comments on the Quality of English LanguageModerate editing of English language is required.
Author Response
Dear reviewer, thanks for your suggestions. We have made the changes you requested.
We proceeded to rewrite the abstract trying to make it more in line with the content of the text.
Figure 1 was created only to schematize what is expressed in the text.
We have revised the section on aminosalicylates, renaming the paragraph to “Aminosalicylates” describing what this group of drugs includes.
Reviewer 4 Report
Comments and Suggestions for Authors
This is a well written overview about liver injury induced by drugs used for therapy of IBD. Apart from small errors in the text, this overview may be supplemented by data about new biologicals and about "small molecules".
I have some suggestions: Lanes 40 - 43. The different ypes of liver damage should be better described. "The indrect from is due to.." Difficult to understand. Idiosyncratic: Dose-dependent or non-dose-dependent? Lane 37. Biochemical parameters, not "biohumeral parameters".
Paragraph Immunosuppressants
I suggest to include a scheme of metabolism of thiopurines. Thus, the complicated text may be better understandable. Some remarks about the differences azathiopurine - mercaptopurine AND - the drug "thioguanine", which might have some advantages.
MTX is correctly discussed. What about cyclosporine A, which is mainly used only in UC. The same für tacrolimus.
The paragraph about anti-integrin-antibodies is very short an could be more detailed.
Antibodies to interleukines (IL12/IL23)?
Newer small molecules shuch as JAK-inhibitors and the new S1P-modulators
Author Response
Dear reviewer, thanks for your suggestions. We have made the changes you requested.
We have tried to better express the characteristics of the various forms of DILI, defining those as dose-dependent and those as dose-independent. Furthermore, as you requested, we have replaced "biohumoral parameters" with "biochemical parameters" in line 37.
We proceeded to create a scheme (figure 2) on the metabolism of thiopurine drugs.
We discussed the paragraph on immunosuppressants, also dealing with cyclosporin A and tacrolimus. We did the same with regards to the paragraph on biological drugs. In fact, we have described in more depth the anti-integrin antibodies, anti-IL12/IL23 antibodies, JAK-inhibitors and S1P modulators.
Reviewer 5 Report
Comments and Suggestions for Authors
The literature review, "Inflammatory bowel disease therapies and acute liver injury" is a valuable study that brings forth pertinent information from the specialized literature. However, certain modifications are warranted:
1. A materials and methods chapter should distinctly present and elaborate on the research strategy employed in the study.
2. An additional chapter focusing on future perspectives in this research area should be included. Recent articles address topics of interest in the presented theme, specifically – implications related to microRNAs (more details can be found in the article https://doi.org/10.3390/biomedicines11072058) and, of course, microbiota transfer as a potential treatment (more details and relevant articles can be found at https://doi.org/10.3390/biomedicines11112930).
3. The discussion chapter is notably lacking. Given the comprehensive nature of the study, this section is essential for addressing discrepancies in the literature and emphasizing crucial aspects of the review that require detailed comparison with themes from existing literature.
4. The conclusions chapter needs to be expanded and modified to encompass most of the review's findings.
Author Response
Dear reviewer, thanks for your suggestions. We have made the changes you requested.
We have not created a "materials and methods" chapter as it is neither an original article nor a meta-analysis.
We have cited the two articles you suggested, indicating microRNAs as possible future strategies for early diagnosis of cases of drug-induced hepatotoxicity.
We have tried to discuss the various paragraphs of the review in more depth, also citing more recent articles and newly introduced drugs in the treatment of IBD.
We have almost completely rewritten the "Conclusions" paragraph to try to make it more in line with the content of the text.
Round 2
Reviewer 2 Report
Comments and Suggestions for Authors
It has been roughly modified.
It is recommended to check it again.The word "such as" appears twice in the abstract.
Comments on the Quality of English Language
There is no problem with English language.
Author Response
Dear reviewer, thanks for your suggestions. We have revised the text.
Reviewer 3 Report
Comments and Suggestions for Authors
The revised version of the manuscript deserves to be published.
Author Response
Dear reviewer, thank you for your opinion.
Reviewer 4 Report
Comments and Suggestions for Authors
The authors have addressed the issues. Therefore, the manuscript may be published.
Neverthelesse, I feel that the English style should be improved. Any suggestion of a native speaker?
Author Response
Dear reviewer, thank you for your suggestion. We do not have a native speaker in our institutions. The text was also revised by one of the authors who has frequented eminent English and American scientific institutes for years. We therefore believe that the text with the changes made in the previous revision phase, with your authoritative consent, could still be sufficiently suitable for publication.
Reviewer 5 Report
Comments and Suggestions for Authors
The authors have made the recommended changes.
Author Response

(The authors gave the same response as above.)
